# NMVOC Emissions from Solvents Use in Greece: Monitoring and Assessment

**Ioannis Sebos *** 🄳 **and Leonidas Kallinikos**

School of Chemical Engineering, National Technical University of Athens, 9 Heroon Polytechniou Street, 15780 Zografou, Greece

\* Correspondence: isebos@mail.ntua.gr; Tel.: +30-210-772-3135

**Abstract:** The use of solvents and other volatile organic chemicals is a significant source of Non-Methane Volatile Organic Compounds (NMVOCs) emissions. Due to the wide spectrum of applications of solvents and numerous locations where these occur, the estimation of NMVOCs emissions can be challenging. The aim of this paper is to present the methodological framework used in Greece for the estimation of NMVOCs emissions. It covers processes and products that use solvents and other volatile organic chemicals in several industries, as well as in households. The framework is based both on existing methods found in the literature and on new emission factors developed in order to reflect the mitigation potential of EU Directives and national legislation aiming at the reduction of NMVOCs emissions. The developed framework was used to forecast future NMVOCs emissions and assess the implemented mitigation actions. Results were verified by comparison with solvent emission estimates from the European Solvent Industry Group.

**Keywords:** Non-Methane Volatile Organic Compounds (NMVOC); emissions inventory; solvents and other volatile organic chemicals; advanced tier emission estimation methods

## 1. Introduction

Air pollution refers to the release into the air of substances detrimental to human health and the environment. The World Health Organization data shows that nearly the entire global population (99%) breathes air that exceeds WHO guideline limits and contains high levels of pollutants, with low- and middle-income countries suffering from the highest exposures [1]. In the context of assessing the extent of air pollution, several studies were elaborated to facilitate environmental policy decision-makers by providing sufficient information to guide their decisions [2,3].

This paper focuses on the air pollutants known as non-methane volatile organic compounds (NMVOCs). NMVOCs include a vast number of hydrocarbons where hydrogen atoms may have been partially or fully replaced by other atoms, such as S, N, O and halogens. The majority of NMVOCs are photochemically reactive and many are toxic [4–6]. NMVOCs have a significant role in the formation of secondary air pollutants such as tropospheric ozone ($O_3$) and secondary organic aerosols, which negatively affect living organisms (primarily their respiratory and cardiovascular systems), buildings and agricultural productivity [7–9].

NMVOCs originate both from natural and anthropogenic sources [10]. Natural sources mainly consist of terrestrial plant emissions [11]. Human-induced sources derive mainly from solvent utilization, traffic emissions, industrial production processes, fossil fuel combustion and biomass burning [12–14]. NMVOCs are also emitted during waste management [15].

In Europe, member states are required to prepare and implement a national air pollution control programme, which includes the policy priorities and measures to reduce the emissions of air pollutants, including NMVOCs [16]. In addition, the national energy and climate plans, compiled and implemented in order to meet the EU Member States

energy and climate targets for 2030, contain measures that contribute the reduction of NMVOCs [17]. To enable the development of a comprehensive and effective emission reduction action plan, the availability of a reliable emission inventory, where emissions are correctly allocated between source categories, along with a methodological framework to quantify the effect of abatement measures are an essential prerequisite [18].

The use of solvents and other volatile organic chemicals is listed as a significant source of NMVOCs emissions in all national air pollutant inventories. It involves diverse emission sources including, for example, the general public using any of a wide range of consumer products, but also extending to a number of processes carried out in practically every branch of industry. Due to the wide spectrum of applications of solvents and numerous locations where these occur, inventory compilers face difficulties in estimating NMVOCs emissions, both in terms of deciding on suitable methodologies available in emission handbooks (e.g., the 2019 EEA/EMEP Guidebook), and in terms of collecting suitable activity data. Furthermore, the generic, easy-to-apply Tier 1 methodologies in emission handbooks are related to production volumes or national population, and therefore do not take into account national mitigation policies and/or plant specific abatement technologies [19].

Despite the significance of NMVOCs emissions, there has been limited research in the scientific and other technical literature to enable the development of a reliable inventory of NMVOCs emissions from solvent use in Greece. The objective of this study is to cover this gap, by creating the necessary framework in terms of NMVOCs accounting methodologies, activity data, baseline emissions, assessment of mitigation actions, and estimation of emissions for tracking progress with respect to the national emission reduction targets.

This paper presents a methodological framework for the estimation of NMVOCs emissions, which covers processes and products that use solvents and other volatile organic chemicals in several industries, as well as in households. This framework has been developed and applied for the compilation of the Greek NMVOCs inventory pursuant to the Convention on Long-range Transboundary Air Pollution (LRTAP Convention) and Directive (EU) 2016/2284. The development of this framework was needed to improve the accuracy of the inventory by reflecting the effect of mitigation actions associated with NMVOCs emissions, which was not previously possible when using generic Tier 1 methodologies. Thus, the emission inventory could serve as a tool to monitor the effect of mitigation policies, to identify priority sectors for mitigation and to assess the achievement of emission reduction targets.

The methodological framework is based both on existing methods found in the literature [20,21] and on new emission factors developed in order to reflect the mitigation potential of EU Directives and National Laws aiming at the reduction of NMVOCs emissions, such as the EU Directive 2004/42/EC that sets maximum VOC content limit values for paints, varnishes and other chemicals, the EU Directives 1999/13/EC and 2010/75/EU that require the application of BATs in industrial applications, etc. Emission factors found in the literature [21], which refer to previous years, were extrapolated to more recent years by applying proxies such as annual spending of households for washing and cleaning products, cosmetics, etc. The main source of national production and consumption data for chemicals was the Hellenic Statistical Authority, supplemented with questionnaires obtained under the framework of EU Directives, e.g., 1999/13/EC, 2010/75/EU, etc. The developed framework was verified by comparing it with the solvent emission estimates from the European Solvent Industry Group (ESIG) [22].

## 2. Materials and Methods

The sources of NMVOCs emissions from solvents can be categorized to the activities presented in Table 1 [19]. The sources with the higher contributions in NMVOCs emissions are discussed in the following paragraphs.

**Table 1.** Categorization of sources of NMVOCs emissions from solvents.

| Sources |
| --- |
| 1. Domestic solvent use |
| 2. Road paving with asphalt |
| 3. Asphalt roofing |
| 4. Coating applications |
| 5. Degreasing, a process for cleaning products from water-insoluble substances such as grease, fats, oils waxes, carbon deposits, fluxes and tars |
| 6. Dry cleaning, any process to remove contamination from furs, leather, down leathers, textiles or other objects made of fibers using organic solvents |
| 7. Chemical products |
| 8. Printing |
| 9. Other solvent and product use |
| 10. Pulp and paper industry |
| 11. Food and beverages industry |

### 2.1. Domestic Solvent Use

Non-Methane Volatile Organic Compounds are used as solvents in a large number of products sold for domestic use. For example, in aerosols NMVOCs such as butane and propane are used as propellants. The products included in this category are divided into a number of categories as presented in Table 2 [19].

**Table 2.** Main categories with regard to the domestic use of solvents.

| Category | Description |
| --- | --- |
| Cosmetics and toiletries | Products for the maintenance or improvement of personal appearance, health or hygiene |
| Household products | Products used to maintain or improve the appearance of household durables |
| Construction/DIY | Products used to improve the appearance or the structure of buildings such as adhesives and paint remover. This sector would also normally include coatings; however, these products fall outside the scope of this section and are therefore omitted |
| Car care products | Products used for improving the appearance of vehicles to maintain vehicles, or winter products such as antifreeze |
| Pesticides | Pesticides, such as garden fungicides, herbicides and insecticides, and household insecticide sprays may be considered as consumer products. Most agrochemicals, however, are produced for agricultural use and fall outside the scope of this section |

To estimate NMVOCs emissions associated with domestic solvent use, data from [21] have been utilized. As described in this study, NMVOCs emissions result from the use of the following products: (a) washing and cleaning products, (b) personal healthcare products, (c) cosmetics, (d) homeware and DIY products and (e) car care products.

The NMVOCs emission factor (EF) from washing and cleaning products (W/C) varies between 0.005 and 0.622 g/capita/year; from personal healthcare products (H/C), between 0.0 and 1.418 g/capita/year; from cosmetics, between 0.333 and 3.432 g/capita/year; from homeware and DIY products, between 0.368 and 1.380 g/capita/year, and from car care products between 0.441 and 1.720 g/cap/year for the period 1996–2006 [21]. For the years after 2006, the EFs are extrapolated by using the annual national consumption of these products as a driver. Annual national consumption data were obtained from EUROSTAT and the Hellenic Statistical Authority. The NMVOCs emissions are calculated by applying the following equation:

$$NMVOCemissions = \sum_i (population \cdot EF_i) \tag{1}$$

where

$EF_i$: Emission factor in g/capita/year of each one of the categories of Table 2.

### 2.2. Coating Applications

This category includes emissions associated with the application of paints for decorative, industrial or other purposes (including lacquers and varnishes and excluding glues and adhesives). The applications are presented in Table 3 [19].

**Table 3.** Main categories with regard to the domestic use of solvents.

| Category | Description |
|---|---|
| Coating application for decorative purposes | Construction, buildings, and domestic use |
| Industrial coating applications | Manufacture of automobiles, car repairing, coil coating, boat building, wood and other industrial uses |
| Other applications of coatings | Application of high-performance protective anti-corrosive materials, fire-resistant coatings to buildings and other large metallic structures and coatings for concrete and road marking |

The NMVOC emissions associated with the annual sales of different types of coatings in Greece were estimated by applying the following equation:

$$\text{NMVOCemissions} = \sum_{j=1}^{CA}\left(\sum_{i=1}^{CT} AD_i P_{ij} EF_i\right) \qquad (2)$$

where

$AD_i$: Activity data of coating type (i) (source: PRODCOM database of the Hellenic Statistical Authority).

$P_{ij}$: Share (%) of coating type (i) that was consumed in each one of the three coating applications (j = decorative, industrial and other) (Table 4).

$EF_i$: Emission factor. For the period 1990–2006, the emission factor of each of the three coating applications was obtained from EMEP/EEA air pollutant emission inventory guidebook 2019 [19]. Since 2007, it is considered that the provisions of the Directive 2004/42/EC on the limitation of emissions of volatile organic compounds due to the use of organic solvents in certain paints and varnishes and vehicle refinishing products have been fully implemented in Greece. The Directive sets maximum VOC content limit values for paints and varnishes, which were implemented in two phases: phase I from 1 January 2007 and phase 2 from 1 January 2010. The EFs were estimated based on VOC content of products.

CT: Coating type.

CA: Coating application.

**Table 4.** Percentage of paint use on coating applications.

| Type of Paint | Percentage of Paint Use | | |
|---|---|---|---|
| | Decorative app. | Industrial app. | Other app. |
| Paints and varnishes, based on acrylic or vinyl polymers dispersed/dissolved in non-aqueous medium, >50% *w/w* | 0.05 | 0.85 | 0.10 |
| Paints and varnishes, based on acrylic or vinyl polymers dispersed or dissolved in an aqueous medium (including enamels and lacquers) | 0.40 | 0.40 | 0.20 |

**Table 4.** *Cont.*

| Type of Paint | Percentage of Paint Use | | |
|---|---|---|---|
| | Decorative app. | Industrial app. | Other app. |
| Paints and varnishes, based on polyesters dispersed/dissolved in a non-aqueous medium, >50% *w/w* | 0.05 | 0.85 | 0.10 |
| Paints and varnishes, based on polyesters dissolved in a non-aqueous medium including enamels and lacquers excluding > 50% *w/w* | 0.05 | 0.85 | 0.10 |
| Other paints and varnishes based on acrylic or vinyl polymers | 0.05 | 0.85 | 0.10 |
| Paints and varnishes: solutions n.e.c. | 0.20 | 0.40 | 0.40 |
| Other paints and varnishes based on synthetic polymers n.e.c. | 0.20 | 0.40 | 0.40 |
| Other paints, varnishes dispersed or dissolved in an aqueous medium | 0.40 | 0.40 | 0.20 |
| Painters fillings | 0.05 | 0.80 | 0.15 |

*2.3. Chemical Products*

This category consists of the emissions from the use of chemical products. This includes many processes such as polysterene foam processing, polyurethane foam processing, polyvinylchloride processing, glues manufacturing, paints manufacturing, inks manufacturing, asphalt blowing (roofing materials), rubber processing and shoes manufacturing.

NMVOCs are mainly estimated by applying the Tier 2 technology-specific method included in [19]. Especially for glues manufacturing, paints manufacturing and inks, the EFs are based on plant-specific data, which were made available through questionnaires obtained under the framework of EU Directives (e.g., 1999/13/EC). The EFs applied range from 8 g/kg product to 11 g/kg product.

Activity data derived from:

- The PRODCOM database of the Hellenic Statistical Authority for asphalt blowing, paints and ink manufacturing;
- The EUROSTAT database for polyurethane foam processing, polysterene foam processing, polyvinylchloride processing, glues and adhesive manufacturing, paints and ink manufacturing.

*2.4. Printing*

This category is related to the emissions from the printing industry. The emissions of NMVOCs associated with the annual ink consumption in Greece are calculated by applying emission factors based on plant-specific data, which were made available through questionnaires obtained under the framework of EU Directives (e.g., 1999/13/EC). The EF 500 g/kg ink was used for the period up to 2008 with a linear decrease up to 219 g/kg ink for the period 2009–2020. This decrease is attributed to improvements of abatement technologies utilized by the printing industries aiming on the reduction of NMVOC emissions. Activity data derive from the PRODCOM database of the Hellenic Statistical Authority.

*2.5. Other Solvent Use*

The following product groups and processes are taken into consideration under this category:

- Fat, edible and non-edible oil extraction;
- Application of glues and adhesives;
- Preservation of wood.

### 2.5.1. Fat, Edible and Non-Edible Oil Extraction

The NMVOCs emissions associated with oil extraction were calculated by applying the Tier 2 technology-specific method included in [19]. Activity data derive from the FAOSTAT database. Abatement efficiencies are assumed to be approximately 80%.

### 2.5.2. Application of Glues and Adhesives

The NMVOC emissions associated with the application of adhesives were calculated by applying the Tier 2 technology-specific method included in [19]. Abatement efficiencies are assumed to range between 76% to 91%. Activity data derive from the PRODCOM database of the Hellenic Statistical Authority.

### 2.5.3. Preservation of Wood

The NMVOCs emissions associated with wood preservation were calculated by applying the Tier 2 technology-specific method included in [19]. The abatement efficiency is assumed to be approximately 37%. Activity data derive from the FAOSTAT database.

### *2.6. Food and Beverages Industry*

This category presents the emissions from food and beverages manufacturing industry (except of vegetable oil extraction). The following products are taken into consideration under this category: meat (including poultry), fish, sugar, margarine and solid cooking fats, bread, biscuits, animal feeds, coffee beans (roast), wine, beer and spirits.

The emissions of NMVOCs associated with these products are calculated by applying the technology-specific method included in [19]. Abatement efficiencies have been assumed to be 90%. Activity data derive from the PRODCOM database of the Hellenic Statistical Authority.

## 3. Results and Discussion

In the following figures, a comparison is made between two datasets. The fist dataset is the time-series of the NMVOCs emissions estimated by applying the methodological framework for Greece described in this paper. The second data set consists of the NMVOCs emissions obtained by applying a generic Tier 1 method from the EMEP/EEA air pollutant emission inventory guidebook 2019 [19].

A Tier 1 is a method that uses readily available statistical data on the intensity of processes (activity rates) and default emission factors. These emission factors assume a linear relation between the intensity of the process and the resulting emissions. The Tier 1 default emission factors also assume an average or typical process description. This method is the simplest method, has the highest level of uncertainty and should not be used to estimate emissions from significant source categories. On the other hand, the country-specific methodologies could be of Tier 2 or Tier 3 level. They are based on more specific emission factors developed on the basis of knowledge of the types of processes and specific technologies and conditions that apply within the country for which the inventory is developed. The Tier 3 level corresponds to methods that are more detailed and more complex compared to the Tier 2 level [19].

As shown in Figure 1, the time series of NMVOCs emissions from domestic solvent use is characterized by significant annual variation within the period 2000–2015, which is attributed to the annual variation of the consumption of relevant products. In contrast the emissions calculated based on the generic Tier 1 approach cannot follow this trend, since the generic approach is a simple EF based solely on the country's population.

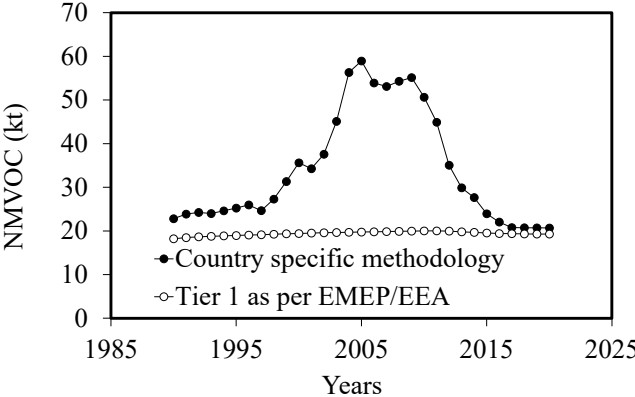

**Figure 1.** NMVOC emissions from domestic solvent use.

Figure 2 presents NMVOCs emissions from coating applications. As shown, over the period 2007–2020, a significant decrease in emissions is observed due to the implementation of Directive 2004/42/CE. The purpose of this Directive was to limit the total content of VOCs in certain paints and varnishes, and it was implemented in two phases: phase I from 1 January 2007 and phase 2 from 1 January 2010. As illustrated in the graph, the decreasing trend in emissions is due to the limits of the VOC content of paints could not be reflected by the generic Tier 1 method.

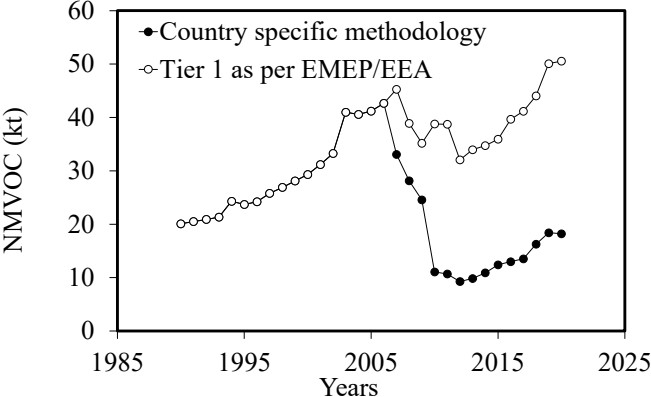

**Figure 2.** NMVOC emissions from coating applications.

The time-series of NMVOCs emissions from chemical products is presented in Figure 3. The most significant deviation between the two datasets occurred for the period 2008–2020. This is attributed to the fact that the methodological framework used for dataset 1 reflects plant-specific data and abatement efficiencies, as discussed in Section 2.3.

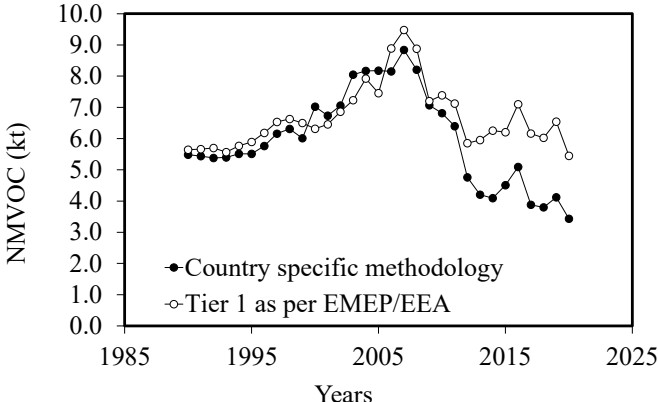

**Figure 3.** NMVOCs emissions from chemical products.

Figure 4 shows NMVOC emissions from printing activities. The difference of the two datasets is also attributed to the use of plant specific data and abatement efficiencies, as discussed in Section 2.4.

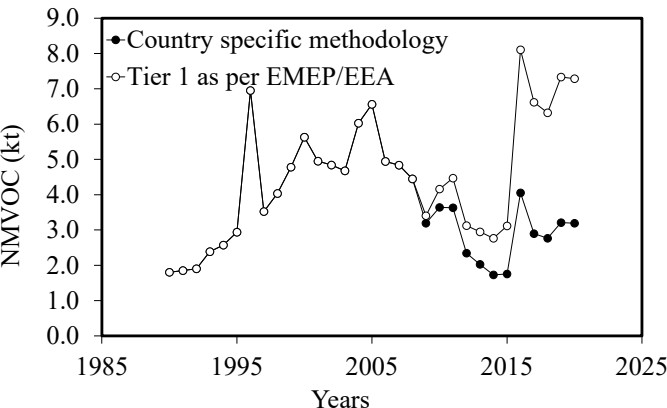

**Figure 4.** NMVOCs emissions from printing.

Figures 5 and 6 present NMVOCs emissions from other solvent use and from the food and beverages industry, respectively. For both categories, the emissions calculated by the methodological framework described in this paper are significantly lower compared to the generic Tier 1 approach. This is because the framework developed includes more accurate and technology specific methodologies, which better model the emissions per solvent type and process. In addition, abatement efficiencies of all relevant process were included in the modelling of emissions.

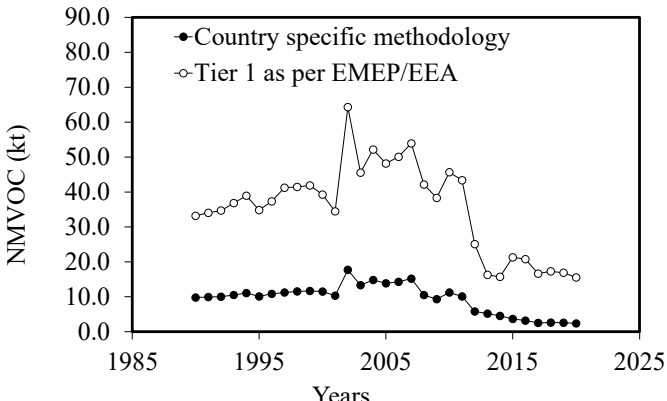

**Figure 5.** NMVOCs emissions from other solvent use.

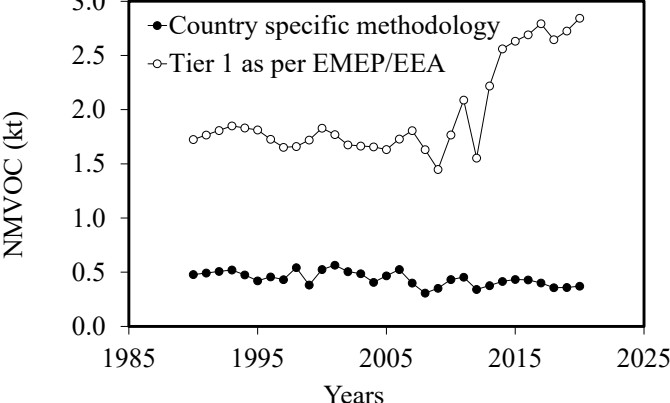

**Figure 6.** NMVOCs emissions from the food and beverages industry.

The total NMVOCs emissions from solvent use in Greece are presented in Figure 7, estimated by applying both the country specific methodological framework and Tier 1. It should be noted that for the categories not described in Section 2 of this paper, for both datasets, the emissions were estimated using the Tier 1 method. The difference between the two datasets represents, to a great extent, the effect of mitigation policies and measures in the solvents category. These comprised maximum VOC content limits of products, BATs, modification of industrial practices for the decrease of NMVOC emissions, installation of abatement technologies, etc.

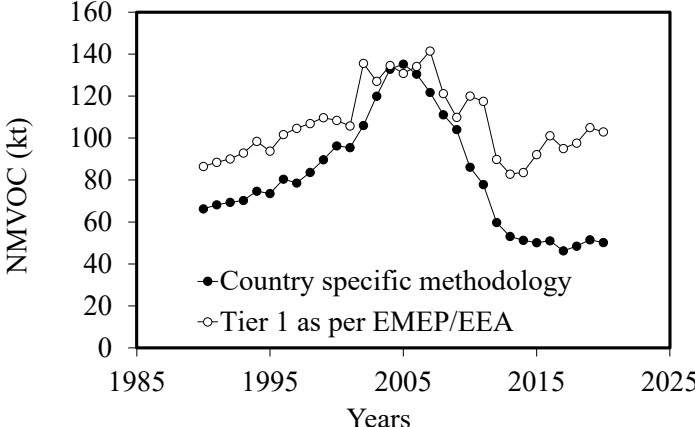

**Figure 7.** NMVOC emissions for the total categories, as per EMEP/EEA, for Greece.

Summarizing, the differences of the NMVOC emission estimations between the country-specific methodological framework and the generic Tier 1 method could be attributed to the drawback of the generic Tier 1 methods to reflect the national mitigation policies and/or plant specific abatement technologies. Tier 1 methods provide estimations that are analogous to the production/consumption volumes of chemicals or the national population, without taking into account the details of applied technologies and processes.

For verification purposes, the estimated NMVOCs emissions obtained by applying the methodological framework described were compared with the emission estimations by the European Solvent Industry Group (ESIG) [22]. The comparison is presented in Table 5. It should be noted that, since emissions reported by ESIG refer to Greece and Cyprus (combined), a disaggregation per country has been performed, based on population.

**Table 5.** NMVOC emissions (kt) as per Country Specific Methodologies and ESIG [22].

| Year | Country Specific Methodology | ESIG for Greece + Cyprus | ESIG for Greece (Assumption) |
|------|------------------------------|---------------------------|-------------------------------|
| 2008 | 111.2 | 85.7 | 79.6 |
| 2009 | 104.0 | 75.6 | 70.2 |
| 2013 | 53.1 | 65.2 | 60.5 |
| 2015 | 50.2 | 49.3 | 45.8 |

As can be concluded by this table, the NMVOC emissions estimated by applying the methodological framework presented here and the estimations performed by ESIG [22] are quite close, especially for the years 2013 and 2015. For the previous years, differences may be attributed to the higher uncertainties of activity data and information about abatement technologies of the more distant historical years.

## 4. Conclusions

To develop a country specific methodological framework, the following sources of information can be used: (a) peer-reviewed journal papers and presentations in scientific conferences; (b) industrial emission data reported under the framework of national legislation

and/or other international channels (e.g., EU legislation); (c) threshold values of national, European or international legislation; and (d) other technical reports (e.g., BATs, etc).

As concluded by the analysis presented above, the estimation of Non-Methane Volatile Organic Compounds emissions based on the generic Tier 1 default methodologies does not represent the specific circumstances of each country, and may result in high overestimation as they do not take into account national mitigation policies and/or plant specific abatement technologies. Therefore, in order to use the emission inventory as a tool to monitor the effect of mitigation policies and to identify priority sectors for mitigation, the development of a country-specific methodological framework is required.

According to the current work, NMVOCs emissions from solvents decreased in Greece for the period 2000–2020 by about 48%, from 96.30kt in 2000 to 50.21 kt in 2020. The most significant source categories of NMVOCs from solvents are coating applications and domestic solvent use, which are responsible for 36% and 42%, respectively, of total NMVOCs emissions from solvents in 2020. The reduction of emissions for the former category can be attributed to the implementation of the Directive 2004/42/CE, which aimed to limit the total content of VOCs in certain paints and varnishes; and for the latter category to the reduction of consumption, particularly after 2008 (economic recession). Emission reductions occurred in the other categories of solvents, too, due to the implementation of the Directives 1999/13/EC and 2010/75/EU and the use of abatement technologies and BATs in the industrial sector.

The NMVOCs emissions are projected to remain at the 2020 levels for the years 2021–2025, as the activity level of solvent related applications and the relevant mitigation policies are not expected to change. A small increase is expected after 2025, due to an increase in solvent using activities. To reduce further emissions in the future, specific mitigation actions targeting the biggest source categories, i.e., coating applications and domestic solvent use, need to be designed and implemented. These actions should be related to additional limitation of the total content of VOCs in paints, varnishes and other products used by households and industry; and/or increase of the utilization of VOC abatement technologies in industrial coating applications.

The methodological framework presented in this paper was verified by comparing the NMVOCs estimations with those by the European Solvent Industry Group (ESIG) [22]. The two datasets agree quite well, especially for the most recent years of the time-series.

**Author Contributions:** Conceptualization, I.S.; methodology, I.S. and L.K.; validation, I.S. and L.K.; formal analysis, L.K.; data curation, L.K.; writing—original draft preparation, I.S. and L.K.; writing—review and editing, I.S.; visualization, L.K. All authors have read and agreed to the published version of the manuscript.

**Funding:** This research received no external funding.

**Institutional Review Board Statement:** Not applicable.

**Informed Consent Statement:** Not applicable.

**Data Availability Statement:** Not applicable.

**Conflicts of Interest:** The authors declare no conflict of interest.

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
