# Peer review of "NMVOC Emissions from Solvents Use in Greece: Monitoring and Assessment"

_atmosphere, doi:10.3390/atmos14010024_

Round 1

Reviewer 1 Report

This is a precise piece of work and I comment the authors for exerting so much effort for their study. This work presents integrated research of NMVOC emissions from solvents use in Greece. The research was interesting, logical and substantial, but some improvements should be made to make it innovative, because solvents use was listed as a significant source of NMVOCs emissions in all national air pollutant inventories.

The keywords and abstract section are reasonable and describe the purpose, methods or procedures, significant new results, and implications.

I have few comments and recommendations:

Line 27, the references the author referred to in this sentence for illustrating the definition are too old.

Line 30, maybe there are more uploading research articles can prove the toxicity of these NMVOCs.

Line 115, I suggest that you can list the formulations of NMVOCs emissions to make it more vivid like line 134.

The conclusion need to be revised to be more concise and focused.

The reference need to be updated to be more dependable.

Some of database resource need to list the website in the reference section.

Author Response

Statement 1: - “This is a precise piece of work and I comment the authors for exerting so much effort for their study. This work presents integrated research of NMVOC emissions from solvents use in Greece. The research was interesting, logical and substantial, but some improvements should be made to make it innovative, because solvents use was listed as a significant source of NMVOCs emissions in all national air pollutant inventories.

The keywords and abstract section are reasonable and describe the purpose, methods or procedures, significant new results, and implications.”

Our response to Statement 1:

We are grateful to the reviewer for the acknowledgment regarding our efforts to prepare an interesting, logical and substantial research study.

The introductory section was modified in order to better reveal the novelty of our research. As we explained in the introduction, although the significance of NMVOCs emissions associated with solvents, there was a lack of comprehensive studies in the scientific and other technical literature to enable the development of a reliable NMVOCs emission inventory in Greece, where emissions are correctly allocated between source categories and mitigation policies and measures are sufficiently reflected in the inventory. The generic, easy-to-apply Tier 1 methodologies from emission handbooks, which were used before our study in Greece, were related to production volumes or national population, and therefore did not take into account national mitigation policies and/or plant specific abatement technologies.

The objective of our study is to cover the above mentioned gap for the solvent sector in Greece, by creating the necessary framework in terms of NMVOC accounting methodologies, activity data, baseline emissions, assessment of mitigation actions, and estimation of emissions for tracking progress with respect to the national emission reduction targets.

To highlight better the contribution and novelty of the paper, lines 66-72 were added in the introductory section.

Statement 2: - “Line 27, the references the author referred to in this sentence for illustrating the definition are too old.”

Our response to Statement 2:

We thank the reviewer for the suggestion. Following his/her guidance, we have included more recent references.

Statement 3: - “Line 30, maybe there are more uploading research articles can prove the toxicity of these NMVOCs.“

Our response to Statement 3:

We thank the reviewer for the suggestion. Following his/her guidance, we have included more recent references.

Statement 4: - “Line 115, I suggest that you can list the formulations of NMVOCs emissions to make it more vivid like line 134.”

Our response to Statement 3:

We thank the reviewer for the suggestion. An equation that describes the calculation of NMVOC emissions was added.

Statement 5: - “The conclusion need to be revised to be more concise and focused.”

Our response to Statement 5:

We thank the reviewer for the suggestion. Please refer to lines 306-325 of the conclusions section, where more text was added to further discuss the evolution of NMVOCs emissions and the drivers behind it. In addition, information about projection of emissions in the future was added.

Statement 6: - “The reference need to be updated to be more dependable. “

Our response to Statement 6:

We thank the reviewer for the suggestion. Following his/her guidance, we have updated the list of references.

Statement 7: - “Some of database resource need to list the website in the reference section.”

Our response to Statement 7:

We thank the reviewer for the suggestion. Following his/her guidance, we have included the missing website addresses (e.g. references 1, 16 and 19).

Reviewer 2 Report

The manuscript presents the emissions calculation for NMVOC from the use of solvent in Greece. The manuscript does not present any novelty, does not compare emission factors to literature neither emissions to the total NMVOC emissions in Greece, does not present how emission factors from industries were determined and validated, etc. It compares emissions to Tier 1 and shows that Tier 2 and country specific methods are more accurate which is trivial, and compares to ESIG emissions for some years. Therefore, I recommend the rejection of the manuscript in its current form. 

Author Response

Statement 1: “The manuscript presents the emissions calculation for NMVOC from the use of solvent in Greece. The manuscript does not present any novelty, does not compare emission factors to literature neither emissions to the total NMVOC emissions in Greece, does not present how emission factors from industries were determined and validated, etc. It compares emissions to Tier 1 and shows that Tier 2 and country specific methods are more accurate which is trivial, and compares to ESIG emissions for some years. Therefore, I recommend the rejection of the manuscript in its current form”

Our response to Statement 1:

We thank the reviewer for the comments. Below, we provide our arguments for each one of the comments.

“The manuscript does not present any novelty”

The introductory section was modified in order to better reveal the novelty of our research. As we explained in the introduction, although the significance of NMVOCs emissions associated with solvents, there was a lack of comprehensive studies in the scientific and other technical literature to enable the development of a reliable NMVOCs emission inventory in Greece, where emissions are correctly allocated between source categories and mitigation policies and measures are sufficiently reflected in the inventory. The generic, easy-to-apply Tier 1 methodologies from emission handbooks, which were used before our study in Greece, were related to production volumes or national population, and therefore did not take into account national mitigation policies and/or plant specific abatement technologies.

The objective of our study is to cover the above mentioned gap for the solvent sector in Greece, by creating the necessary framework in terms of NMVOC accounting methodologies, activity data, baseline emissions, assessment of mitigation actions, and estimation of emissions for tracking progress with respect to the national emission reduction targets.

To highlight better the contribution and novelty of the paper, lines 66-72 were added in the introductory section.

“The manuscript … does not compare emission factors to literature”

We have compared our results with scientific literature. As we have highlighted in the abstract, the results of the developed framework were verified by comparison with solvent emission estimates from the European Solvent Industry Group. The ESIG estimates have been published at “Pearson, J. European solvent VOC emission inventories based on industry-wide information. Atmos. Environ. 2019, 204, 118-124.”

Please refer to lines 280-291 of the manuscript and Table 5.

“The manuscript… does not compare …neither emissions to the total NMVOC emissions in Greece”

We explained in the introduction section that the manuscript focuses on NMVOC emissions from solvent sector, and not the total NMVOC emissions in Greece (including all source categories), because solvent sector is listed as one of the major sources of NMVOCs and due to the challenges and difficulties that inventory compilers face in estimating NMVOCs emissions, both in terms of deciding on suitable methodologies available in emission handbooks, and in terms of collecting suitable activity data. Please refer to lines 54-65 of the manuscript.

“The manuscript … does not present how emission factors from industries were determined and validated”

We included in the manuscript the sources of all data and information, which we have used, to estimate EFs and emissions. As the comment is about industry, relevant information can be found in sections 2.2 – 2.6. For example, for coating applications we reflected in the EFs the provisions of the Directive 2004/42/EC on the limitation of emissions of volatile organic compounds due to the use of organic solvents in certain paints and varnishes. Details about how we modeled the mitigation effect of the Directive are provided in section 2.2 of the manuscript (lines 127-153). Similarly, for the other industrial sectors, we used facility-level data to estimate EFs. Please refer to sections 2.3, 2.4, etc.

“It compares emissions to Tier 1 and shows that Tier 2 and country specific methods are more accurate which is trivial”

The objective of the paper is not to show whether an advanced Tier method is better compared to Tier 1. An advanced Tier method is more accurate only when it is well developed and verified. Therefore, it could be worse than Tier 1, if it is poorly developed. And you need verification to be sure whether the new estimates are more accurate.

This was the purpose of our study. To prepare a methodological framework that contains advanced Tier methods and use facility level data, in order to improve the accuracy of the NMVOCs inventory of solvents sector. To enable the development of a reliable inventory of NMVOCs emissions from solvents use in Greece. To create the necessary framework in terms of NMVOCs accounting methodologies, activity data, baseline emissions, assessment of mitigation actions, and estimation of emissions for tracking progress with respect to the national emission reduction targets.

To ensure that the new framework provides better, i.e. more accurate, results compared to the generic Tier 1 method, we verified our results with the estimates of ESIG, which have been published in the scientific literature.      

 “and compares to ESIG emissions for some years”

The same methods and data sources are used for all years of the time-series. Therefore, we consider that the verification through the comparison of the emission estimates for 4 years is sufficient.

Reviewer 3 Report

The authors present a methodological framework for estimating non-methane volatile organic compound emissions in Greece. For better facilitate the reader, I suggest that the author could make the following changes.

1.The contribution and novelty of the paper should be described in the introduction.

2. Authors must strengthen the literature review section. In the introduction, the authors clearly do not have enough information about the existing studies, and it is suggested that the authors must examine more PM2.5 studies done by others. See for example “PM2.5 volatility prediction by XGBoost-MLP based on GARCH models”and"Spatio-Temporal Characteristics of PM2.5 Concentrations in China Based on Multiple Sources of Data and LUR-GBM during 2016–2021".

3. Please describe lines 82-96 in tabular form. The same goes for 123-129. Also the author needs to elaborate on the data.

4. The author mentions”According to the current work, Non-Methane Volatile Organic Compounds emissions from Solvents decreased in Greece for the period 2000-2020 by about 48%.” Can you make a before and after comparison?

5. Can the authors make a projection of NMVOCs emissions for 2021-2025? The results section needs more discussion of recommendations and policies.

Author Response

Statement 1: “The contribution and novelty of the paper should be described in the introduction.”

Our response to Statement 1:

We thank the reviewer for the suggestion. The introductory section was modified in order to better reveal the novelty of our research. As we explained in the introduction, although the significance of NMVOCs emissions associated with solvents, there was a lack of comprehensive studies in the scientific and other technical literature to enable the development of a reliable NMVOCs emission inventory in Greece, where emissions are correctly allocated between source categories and mitigation policies and measures are sufficiently reflected in the inventory. The generic, easy-to-apply Tier 1 methodologies from emission handbooks, which were used before our study in Greece, were related to production volumes or national population, and therefore did not take into account national mitigation policies and/or plant specific abatement technologies.

The objective of our study is to cover the above mentioned gap for the solvent sector in Greece, by creating the necessary framework in terms of NMVOC accounting methodologies, activity data, baseline emissions, assessment of mitigation actions, and estimation of emissions for tracking progress with respect to the national emission reduction targets.

To highlight better the contribution and novelty of the paper, lines 66-72 were added in the introductory section.

Statement 2: “Authors must strengthen the literature review section. In the introduction, the authors clearly do not have enough information about the existing studies, and it is suggested that the authors must examine more PM2.5 studies done by others. See for example “PM2.5 volatility prediction by XGBoost-MLP based on GARCH models”and"Spatio-Temporal Characteristics of PM2.5 Concentrations in China Based on Multiple Sources of Data and LUR-GBM during 2016–2021"

Our response to Statement 2:

We thank the reviewer for the suggestion. To acknowledge the current efforts made and existing studies to investigate the impact of air pollution on public health and environment, we have included the following text in the Introduction, line 24 of the manuscript (“Air pollution refers to the release of pollutants into the air that are detrimental to human health and environment. WHO data show that almost all of the global population (99%) breathe air that exceeds WHO guideline limits and contains high levels of pollutants, with low- and middle-income countries suffering from the highest exposures [1]. In the context of assessing the extent of air pollution, several studies were elaborated to facilitate environment policy decision-makers by providing sufficient information to guide their decisions [2, 3].”

Statement 3: Please describe lines 82-96 in tabular form. The same goes for 123-129. Also the author needs to elaborate on the data.

Our response to Statement 3:

We thank the reviewer for the suggestion. Following his/her guidance, we have included two new tables in the manuscript.

Statement 4: “The author mentions ”According to the current work, Non-Methane Volatile Organic Compounds emissions from Solvents decreased in Greece for the period 2000-2020 by about 48%.” Can you make a before and after comparison?”

Our response to Statement 4:

We thank the reviewer for the suggestion. More information was added about the change of emissions between 2000-2020 and the drivers behind it. Please refer to lines 306-316 of the manuscript.

Statement 5: “Can the authors make a projection of NMVOCs emissions for 2021-2025? The results section needs more discussion of recommendations and policies”

Our response to Statement 5:

We thank the reviewer for the suggestion. Information about projections of emissions in the future and the required mitigation policies to achieve further reduction of emissions was added to the text. Please refer to lines 317-325 of the manuscript.

Round 2

Reviewer 2 Report

The authors have improved the manuscript and presented more clearly the concerns I have presented. I recommend the publication of this version in its current improved form. 

Reviewer 3 Report

I think the author has made the required corrections and recommend accepting.